# AstRL: Analog and Mixed-Signal Circuit Synthesis with Deep Reinforcement Learning

## Abstract

Analog and mixed-signal (AMS) integrated circuits (ICs) lie at the core of modern computing and communications systems. However, despite the continued rise in design complexity, advances in AMS automation remain limited. This reflects the central challenge in developing a generalized optimization method applicable across diverse circuit design spaces, many of which are distinct, constrained, and non-differentiable. To address this, our work casts circuit design as a graph generation problem and introduces a novel method of **A**MS **S**yn**T**hesis driven by deep **R**einforcement **L**earning (**AstRL**). Based on a policy-gradient approach, AstRL generates circuits directly optimized for user-specified targets within a simulator-embedded environment that provides ground-truth feedback during training. Through behavioral-cloning and discriminator-based similarity rewards, our method demonstrates, for the first time, an expert-aligned paradigm for generalized circuit generation validated in simulation. Importantly, the proposed approach operates at the level of individual transistors, enabling highly expressive, fine-grained topology generation. Strong inductive biases encoded in the action space and environment further drive structurally consistent and valid generation. Experimental results for three realistic design tasks illustrate substantial improvements in conventional design metrics over state-of-the-art baselines, with 100% of generated designs being structurally correct and over 90% demonstrating required functionality.

## 1 Introduction

Modern application-specific integrated circuits (ASICs) have scaled rapidly to meet the increasing computational and data throughput demands of cloud and artificial intelligence (AI) applications. By integrating thousands of integrated circuits at the server scale, leading domain-specific computing systems have demonstrated computational performance in excess of 40 exaFLOPS (Lohmeyer, 2024; Vahdat, 2025). Central to these advancements are complex, high-performance analog and mixed-signal (AMS) circuits that provide critical functions ranging from precision data-conversion to high-speed data serialization/ de-serialization (SERDES) over electrical and optical interconnect.

Situated at the intersection of the analog and digital domains, AMS design encompasses the considerations of both while also inheriting their distinct challenges. Unlike purely digital circuits, which can be simplified with Boolean abstractions to enable logic synthesis (McCluskey Jr., 1956; Brayton et al., 1982; McGeer et al., 1993), AMS circuits operate directly on the continuous, non-linear dynamics of device physics, merging digital and analog techniques to realize complex, high-performance functions. Owing to this complexity, current design processes rely on quasi-linear approximations (Liu & Nagel, 1982; Jespers & Murmann, 2017) and deep domain expertise, which has led to the development of numerous non-standardized design flows. However, in recent years, these methodologies have been significantly challenged by the physical nonidealities and secondary effects introduced in new low-nanometer technology nodes (Nauta, 2024) as well as discrete parameter quantization imposed by fabrication limitations (Loke et al., 2018).

Inspired by key advances in adjacent fields (You et al., 2018a;b), we propose a **generalized method for AMS circuit design** that formulates circuit synthesis as a sequential graph generation problem. Our approach employs a policy-gradient method with a graph backbone that leverages the relational nature of circuit topologies to optimize over challenging design spaces. To extend this approach to

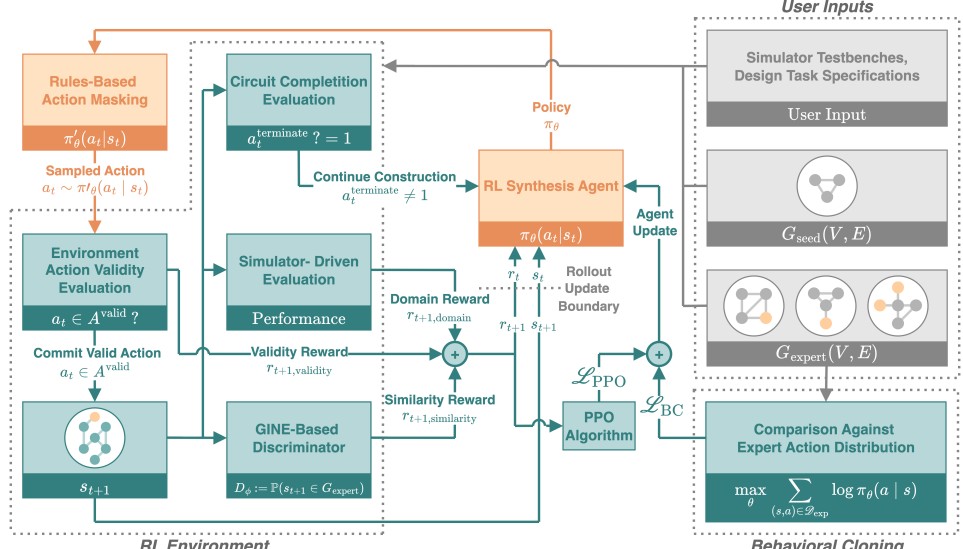

Figure 1: Block diagram of the AstRL design framework.

a broad range of AMS circuits, we further introduce a generalized reward formulation that abstracts domain-specific performance specifications into a unified optimization objective.

A crucial requirement for AMS automation is the integration of topological priors that regularize the search space to preserve standard design conventions. This important condition has limited the adoption of early AMS automation methods (Koza et al., 1997; Meissner & Hedrich, 2015). To address this, our work merges symmetry-aware action space design with (1) an annealed behavioral cloning scheme (Pomerleau, 1988) and (2) a discriminator-based similarity reward. These mechanisms provide important initial expert alignment while also preserving exploration flexibility.

Finally, practical deployment requires extensive verification. In conventional design flows, circuits must be verified with ground-truth simulators, a critical process that is as resource-intensive as the design process itself (Metrans, 2023). Our work embraces this principle by embedding circuit simulators and extensive structural constraints into an RL environment. By introducing ground-truth simulation results and constraints into our training, our framework generates circuit topologies that are both valid by construction and directly optimized for performance.

In summary, this work contributes (1) a generalized policy-gradient-based formulation of AMS design, (2) a broadly applicable reward design, (3) a symmetry-aware action space with expert-alignment for convention-consistent topology generation, and (4) robust metric definitions and validation processes encoded in a comprehensive RL environment. We demonstrate 100% circuit netlisting validity and over 90% simulation validity on experiments, achieving state-of-the-art results.

## 2 BACKGROUND AND RELATED WORK

Broadly, the circuit design process involves constructing topologies using the base components available in a target process technology (e.g., Skywater 130nm, TSMC 16nm, etc.) to meet multiple performance objectives, subject to structural constraints (Gray et al., 2001; Razavi, 2000). Formative techniques like Koza et al. (1997); Meissner & Hedrich (2015) employed evolutionary algorithms, but lacked necessary expert-guidance. Later automation frameworks Crossley et al. (2013); Chang et al. (2018); Guo et al. (2025) utilized generator-based methodologies that formalized design procedures into codified scripts. While effective, these frameworks do not provide algorithmic basis and rely on users to encode extensive sets of design routines for various circuit classes. Ajayi et al. (2020) addressed these limitations by abstracting AMS circuits to work within digital synthesis flows. However, these simplifying abstractions introduced deviations from physical basis, resulting in less performant designs.

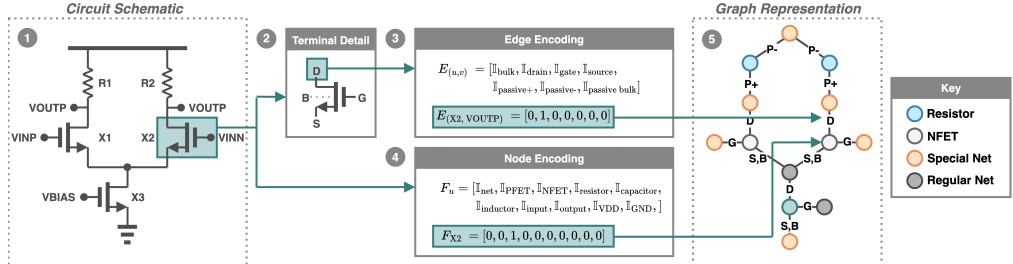

Figure 2: Overview of the graph representation of a circuit topology. (1) Raw circuit schematic for a differential resistive-loaded amplifier; (2) Diagram of a transistor with pins detailed; (3) Edge encoding with definition and translated feature; (4) Node encoding with definition and translated feature; (5) Final graph representation.

More recently, multiple efforts have explored the application of machine learning (ML) techniques for automating circuit design. For digital design, ML has been successfully applied to logic synthesis, where Boolean abstractions provide a natural foundation for learning (Bai et al., 2025; Wang et al., 2024). In contrast, AMS design lacks an analogous discrete abstraction, necessitating different modeling strategies. Foundational approaches (Lai et al., 2024; Chang et al., 2024) rely on extracting representations from large language models (LLMs) through prompt-engineering and text-based circuit representations. Crucially, these approaches are limited by graph sequentialization and the loss of structural properties such as permutation invariance. Moreover, performance is heavily bounded by LLM domain knowledge. Domain-tailored approaches include Dong et al. (2023), which utilized behavioral subcircuits and graph neural networks (GNNs) to generate topologies. The limitations of this inexpressive subcircuit-based construction were subsequently resolved in Gao et al. (2025), which introduced transistor-level, transformer-based circuit generation guided by reinforcement learning with human feedback (RLHF). Critically, none of these existing approaches employ direct ground-truth supervision during model training, limiting both the quality of learned representations and their practical utility in conventional design contexts.

## 3 APPROACH

We formulate circuit topology generation as a sequential graph generation problem framed within a Markov decision process (Sutton & Barto, 2018). At each construction step $t$, state $s_t \in \mathcal{S}$ is represented by a partially constructed circuit topology. To expand the graph, the agent selects an action $a_t \in \mathcal{A}^{\text{valid}}$, where $\mathcal{A}^{\text{valid}}$ is the set of valid actions obtained by masking the full action space $\mathcal{A}$ with domain-specific environment constraints. After addition, rewards are assessed and policy $\pi(a_t \mid s_t)$ is updated. Across training iterations, the policy $\pi(a_t \mid s_t)$ is optimized to maximize the expected return. The complete framework is illustrated in Fig. 1.

### 3.1 GRAPH REPRESENTATION FOR CIRCUIT TOPOLOGIES

AMS circuits are inherently relational structures with function defined by component topology and connectivity. The standard domain representation for circuits is the netlist, a text-based description of components and interconnections. However, this representation is sensitive to naming conventions and lacks permutation invariance, allowing identical circuits to map to different textual forms. To better incorporate structural inductive biases, we instead adopt a graph-based representation. While many such representations are possible, as illustrated in adjacent work (Hakhamaneshi et al., 2023; Dong et al., 2023), we specifically opt for a compact representation similar to Ren et al. (2020) due to a low hop-count suitable for graph-based learning.

As shown in Fig. 2, a circuit is modeled as a graph, $G = (V, E)$ with $d$ node types and $b$ edge types. Each node $u \in V$ is associated with a one-hot feature vector $F_u \in \{0, 1\}^d$. Node types are divided into (1) active components, (2) passive components, (3) generic nets, and (4) special nets. Within the circuit context, **nets** are nodes connecting multiple components. Edge types encode the terminals of the components being connected. Each edge $(u, v) \in E$ is associated with an edge attribute vector $E_{(u,v)} \in \{0, 1\}^b$. Importantly, this representation enables the description of fine-grain actions and

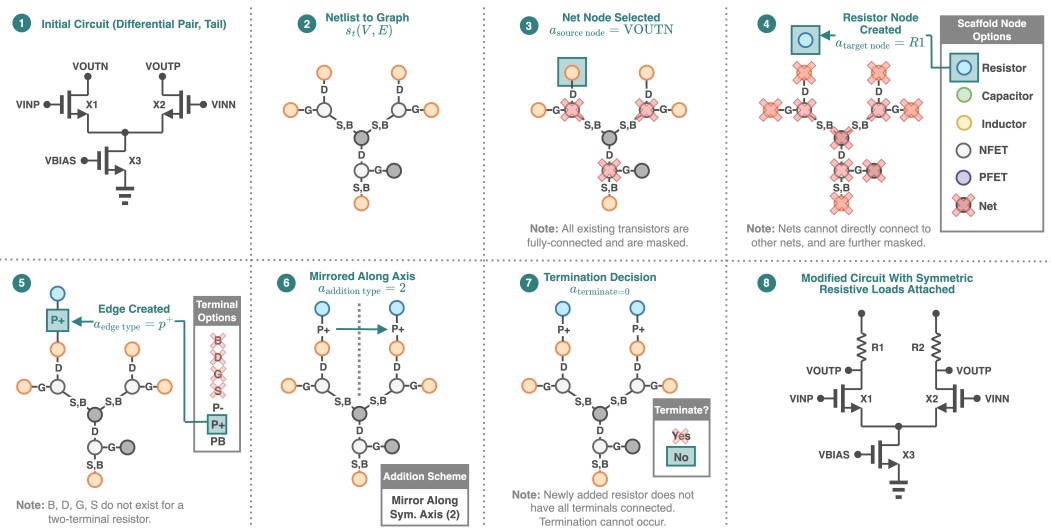

Figure 3: Overview of a single action, with masking. (1) Initial circuit; (2) Netlist converted to graph; (3) Source node selection; (4) Target node selection and addition; (5) Edge terminal selection and addition; (6) Action modification; (7) Termination decision; (8) Modified circuit.

low-level constraints. Furthermore, we note its compatibility with expressive graphical models (Hu* et al., 2020; Yang et al., 2021).

## 3.2 ACTION SPACE ENCODING FOR STRUCTURAL CONSISTENCY

A critical feature for circuit generation is structural consistency and validity. In this section, we detail a fine-grained action schema that samples its components from learned distributions. In particular, we highlight how our action design encodes inductive biases that guarantee valid and consistent construction. Formally, we define an action for partially constructed graph $s_t$ as:

$$a_t = [a_{\text{source node}}, a_{\text{target node}}, a_{\text{edge type}}, a_{\text{addition type}}, a_{\text{terminate}}] \tag{1}$$

$$
\begin{aligned}
X_{s_t} &= \text{GINE}(s_t), \\
a_{\text{source node}} &\sim \text{Softmax}(f_1(X_{s_t})), & a_{\text{source node}} &\in [0, |V|), \\
a_{\text{target node}} &\sim \text{Softmax}(f_2(X_{s_t}, a_{\text{source node}})), & a_{\text{target node}} &\in [0, |V| + d), \\
a_{\text{edge type}} &\sim \text{Softmax}(f_3(X_{s_t}, a_{\text{source node}}, a_{\text{target node}})), & a_{\text{edge type}} &\in [0, b), \\
a_{\text{addition type}} &\sim \text{Softmax}(f_4(X_{s_t}, a_{\text{source node}}, a_{\text{target node}}, a_{\text{edge type}})), & a_{\text{addition type}} &\in [0, O_{\text{types}}), \\
a_{\text{terminate}} &\sim \text{Softmax}(f_5(X_{s_t}, a_{\text{source node}}, a_{\text{target node}}, a_{\text{edge type}})), & a_{\text{terminate}} &\in \{0, 1\}
\end{aligned}
\tag{2}
$$

where $O_{\text{types}}$ denotes the number of symmetric action modifiers, discussed later.

Subvector $[a_{\text{source node}}, a_{\text{target node}}, a_{\text{edge type}}]$ of Eq. 1 details how pair-wise connections are created between nodes. Importantly, the source node $a_{\text{source node}}$ should be chosen from the existing nodes in partial graph $s_t$ to prevent formation of a disconnected subgraph. Target node $a_{\text{target node}}$ can be chosen from existing nodes or added from a set of candidate scaffold nodes, subject to structural rules. As shown in Fig. 3, constraint-violating selections are suppressed with probability masks. If a constraint-violating selection is made, the graph construction will not be updated ($s_{t+1} = s_t$). Appendix A.3 details structural constraints.

Many AMS circuits exhibit a natural symmetry with respect to the axis between the supply and ground nodes. These circuits are called **differential circuits**. Formally, differential circuits define a graph automorphism $\phi : V \to V$ such that:

$$
\begin{aligned}
\phi(u_1) = u_2, \quad &\forall (u_1, u_2) \in \text{symmetric branch pairs} \\
(u, v) \in E &\iff (\phi(u), \phi(v)) \in E
\end{aligned}
\tag{3}
$$

Application of naive graph expansion to circuits containing this symmetry significantly increases trajectory depth. Our framework addresses this with action $a_{\text{addition type}}$ which defines five different symmetric modifiers that transform action subvectors. Appendix A.2 illustrates these modifiers in detail. Specifically, these schemes establish (1) singular common-mode modification, (2) symmetric pair modification, (3) symmetric pair to common-mode component modification, (4) symmetric pair to common-mode net modification, and (5) common-mode to symmetric pair modification. In addition to reducing trajectory depth, $a_{\text{addition type}}$ enables the framework to preserve circuit structural motifs and symmetry during the graph construction process, thereby further ensuring a structurally consistent action space $\mathcal{A} = \mathcal{A}_1 \times \mathcal{A}_2 \times \cdots \times \mathcal{A}_k$.

As noted earlier, $\mathcal{A}$ may contain actions that cannot be taken due to domain-specific structural constraints. As an example, connecting two net nodes induces an invalid electrical short. We therefore introduce masking to reduce the size of the effective action space to subspace $\mathcal{A}^{\text{valid}}$. Let $a_{<k} = [a_0, \ldots, a_{k-1}]$ denote the prefix of length $k$. At each step $k$, not all choices in $\mathcal{A}_k$ are feasible. To enforce validity, we define a binary mask $M_k(s, a_{<k}) \in \{0,1\}^{|\mathcal{A}_k|}$ which prevents probability mass from being assigned to infeasible actions prior to action sampling. For $i = 1, \ldots, |\mathcal{A}_k|$, $M_k(s, a_{<k})_i = 1$ if an action is valid, otherwise 0. The valid action set at each step $k$ can therefore be described as:

$$\mathcal{A}_k^{\text{valid}}(s, a_{<k}) = \{a_{k,i} \in \mathcal{A}_k \mid M_k(s, a_{<k})_i = 1\}. \tag{4}$$

This formulation guarantees that the mask for $a_1$ depends on $a_0$, the mask for $a_2$ depends on $(a_0, a_1)$, and so forth, ensuring that infeasible partial trajectories are eliminated as the sequence is constructed.

### 3.3 GENERALIZED REWARD DESIGN

Reward design for AMS tasks presents two primary challenges. First, different tasks require optimization over distinct sets of multi-objective performance parameters (e.g., gain–bandwidth in amplifiers vs. delay–power trade-offs in comparators). Second, partially constructed graphs are typically nonfunctional and therefore cannot be evaluated for performance-based rewards. Additionally, we note that the design space is itself highly nonlinear and often discontinuous. As such, we define three types of rewards: (1) validity, (2) similarity, and (3) domain-specific performance. Formally, the total reward becomes:

$$r(s_t, a_t) = r_{\text{validity}} + r_{\text{similarity}} + r_{\text{domain}} \tag{5}$$

The validity reward penalizes actions that violate structural consistency. The similarity reward is assessed through a classifier network $D_\phi$ that assigns a small positive return when the partially constructed subgraph is consistent with expert subgraphs and a small negative return otherwise. This both provides expert alignment and encourages trajectory stability in the absence of strong reward signals. Finally, domain-specific rewards are used to (1) broadly encapsulate performance metrics for diverse tasks and (2) reward supply-ground connectivity. Formally, this is defined as:

$$r_{\text{domain}} = r_{\text{sim validity}} + r_{\text{opt}} + r_{\text{opt success}}, \quad \text{where } r_{\text{opt}} = \sum_{p \in P} w_p \cdot r_{\text{opt},p}(v, p) \tag{6}$$

where $P$ is the set of performance specifications, $p$ is a single specification within the set, $v$ is the measured value from simulation, and $w_p$ is a weighting factor. Each specification-level contribution is:

$$r_{\text{opt},p}(v, p) = \begin{cases} 1 - \dfrac{v - p}{e}, & \text{if the objective is to match a target with bound } e, \\[2mm] \dfrac{v - p}{v - 3p}, & \text{if the objective is to minimize,} \\[2mm] \dfrac{v - p}{v + p}, & \text{if the objective is to maximize.} \end{cases} \tag{7}$$

where $e$ is the allowed error bound on performance targets. This formulation ensures that completed circuits are either rewarded for achieving functional validity and proximity to performance targets or for exceeding specification thresholds.

## 3.4 Expert Alignment Via Behavioral Cloning

We utilize Proximal Policy Optimization (PPO) (Schulman et al., 2017) to train our policy using a gradient-based approach. The clipped PPO objective function is defined as:

$$\mathcal{L}_{\pi_\theta}^{\text{PPO}} = \mathbb{E}_t\Big[ \min\Big( \rho_t(\pi_\theta, \pi_{\theta_k})A_t^{\pi_{\theta_k}}, \text{clip}\big(\rho_t(\pi_\theta, \pi_{\theta_k}),\, 1-\epsilon,\, 1+\epsilon\big)A_t^{\pi_{\theta_k}}\Big) + c\,\mathcal{H}[\pi_\theta](s_t)\Big] \quad (8)$$

To improve alignment with expert actions, we augment the primary PPO objective with behavioral cloning (BC) (Pomerleau, 1988). The BC objective function is defined as:

$$\max_\theta \sum\nolimits_{(s,a)\in\mathcal{D}_{\text{expert}}} log\pi_\theta(a|s) \quad (9)$$

To implement the BC objective, we align agent actions with sets of expert trajectories, $\mathcal{D}_{\text{expert}}$. Each expert trajectory $\tau_e = \{(a_0, s_0), (a_1, s_1)...(a_t, s_t)\}$ is a sequence of state-action pairs that incrementally constructs a complete expert design, starting from a sampled subgraph. We pretrain the policy with BC across a diverse set of circuit classes, yielding a strong initialization that captures expert design priors over a broad range of topologies. During the core training phase, BC continues against a small set of experts with gradually annealed influence to encourage exploration.

The final optimization objective is therefore a joint loss that combines the PPO and BC objectives to enable the agent to perform performance-driven exploration using a stable RL core while also retaining structured initial guidance.

$$\theta^* = \max_\theta \mathcal{L}_{\pi_\theta}^{PPO} + \lambda_0\lambda_1^k \sum\nolimits_{(s,a)\in\mathcal{D}_{exp}} log\pi_\theta(a|s) \quad (10)$$

## 4 Results

### 4.1 Experiment Setup

**Dataset and Framework Setup:** We pretrain our policy on a dataset of 1172 circuits, derived from a subset of 3350 designs released in Gao et al. (2025). These circuits satisfy standard supply connectivity.

The policy network consists of a 3-layer GINE (graph isomorphism net with edge features) network with 64-dimension node embeddings, followed by 3-layer fully-connected neural nets with 64-dimension hidden layers for each respective action selection. The discriminator network similarly consists of a 3-layer GINE network with 64-dimension node embeddings, followed by fully-connected layers for discriminator classification. The GINE architecture is chosen for its 1-WL expressivity achieved through edge-conditioned message passing, a property that is particularly important in circuit graphs where both device types and connection types directly influence functionalityXu et al. (2019). This makes GINE a strong fit for modeling circuits, as it provides a standard and expressive framework capable of capturing subtle structural variations. Even small topological changes in circuits, such as a single incorrect or missing connection, can fundamentally alter circuit behavior or lead to complete failure. In total, this model contains approximately 244K parameters.

**Task Descriptions:** Our approach is evaluated on three representative circuit design tasks: (1) a ring oscillator (RO), (2) a comparator, and (3) an operational transconductance amplifier (OTA). Each task features multiple objectives and constraints, and generated topologies range from highly nonlinear circuits with digital outputs (RO, comparator) to classical linear circuits (OTA). To seed each task, minimal substructures of 3-7 transistors are provided, as illustrated in Appendix Figs. 7 - 9. In each task, the agent made unit-sized component additions. For the RO task, the objective was to achieve a 4.75 GHz oscillation frequency and 50% duty cycle with $\pm$ 10% error margin. Expert designs were a 3-stage and 7-stage inverter-based RO. For the comparator task, the objective was to obtain sub-600 ps output delay for a 100fF load and input noise below 0.7 $\text{mV}_{\text{rms}}$. The expert design was a StrongARM comparator Madden & Bowhill (1990). Finally, for the OTA task, the objective was to obtain AC gain above 17 dB, bandwidth below 10 MHz, and a gain difference below 2 dB. Expert designs were a five-transistor (5T) OTA with common-mode feedback resistors, 5T OTA with parallel resistive load, and 5T OTA with series resistive load. All tasks had additional power constraint specifications to limit spurious designs.

These tasks demonstrate our method's ability to (1) target precise specifications (e.g., RO frequency target) and (2) minimize/maximize performance beyond thresholds (e.g., OTA gain). All tasks were verified in simulator-driven testbenches to extract raw performance. Experiments were conducted in the Skywater 130nm process to demonstrate viability with with commercial process technologies.

**Reward Implementation:** We implement our three component reward consisting of validity, similarity, and domain specific rewards as follows:

1. **Validity:** Invalid actions receive $r_{validity} = -2$, otherwise 0.

2. **Similarity:** A task-specific discriminator provides $r_{similarity} \in -1, +1$, trained once per task on expert subgraphs (positives) and pretrained policy rollouts (negatives).

3. **Domain-specific:** Assigned at rollout termination ($a_{terminate} = 1$ or step limit), structurally invalid circuits (e.g., missing supply) incur $r_{sim\ validity} = -2$. Valid but simulation-invalid designs yield $+3$; simulation-valid designs yield $+30$. Performance rewards $r_{opt}$ are applied per-specification with weight $w_p = 15$. An additional success bonus $r_{opt,success} = +10$ if all constraints are met.

**Evaluation Metrics:** For quantitative evaluation of experimental performance, we define four metrics that reflect both topology generation concerns and conventional design considerations.

1. **Netlist Validity:** Percentage of generated designs that map to syntactically correct netlists. Does not require simulation validity or specification fulfillment.

2. **Simulation Validity:** Percentage of generated designs that can be simulated successfully. Testbench evaluations must conclude with nominal results, specification fulfillment is not required.

3. **Specification Fulfillment:** Percentage of generated designs that meet all raw design constraints. No proxy figure of merit (FoMs) are used in this work for assessment.

4. **Novelty:** Percentage of generated designs that are not present in the provided dataset. Following prior works, designs need not fulfill any of the other three defined metrics.

**Baselines:** We evaluate AstRL against openly available, state-of-the-art (SoTA) frameworks for transistor-level circuit generation. The method of Chang et al. (2024) is excluded as it targets only power-converter design, while Dong et al. (2023) is omitted due to its use of behavioral modeling rather than simulations for evaluation. Our baselines therefore include: (1) AnalogCoder (Lai et al., 2024), which utilizes LLMs with domain-specific prompt engineering for circuit generation, and (2) AnalogGenie (Gao et al., 2025), which employs a transformer-based architecture combined with reinforcement learning with human feedback (RLHF) to generate tokenized circuit representations.

| Task | Metric | AstRL (no disc.) | AstRL (no BC) | AstRL (no mask) | AstRL (no sym.) | AstRL* | AnalogCoder (GPT-3.5) | AnalogCoder (GPT-4o) | AnalogCoder (GPT-5†) | AnalogGenie |
|------|--------|---------|---------|---------|---------|--------|-----------|-----------|-----------|-------------|
| RO | Netlisting Validity (%) | 100 | 100 | 100 | 100 | 100 | 53.3 | 83.3 | 80.0 | – |
| | Simulation Validity (%) | 100 | 0.0 | 0.0 | 90.1 | 90.1 | 0.0 | 26.7 | 73.3 | – |
| | Specification Fulfilling (%) | 0.0 | 0.0 | 0.0 | 13.6 | 13.6 | 0.0 | 0.0 | 20.0 | – |
| | Novelty (%) | 6.9 | 100 | 100 | 100 | 100 | 86.7 | 83.3 | 100 | – |
| Comp. | Netlisting Validity (%) | 100 | 100 | 100 | 100 | 100 | 53.3 | 60.0 | 100 | – |
| | Simulation Validity (%) | 8.90 | 0 | 33.1 | 99.2 | 99.2 | 30.0 | 46.7 | 80.0 | – |
| | Specification Fulfilling (%) | 6.08 | 0 | 33.1 | 99.2 | 99.2 | 26.7 | 43.3 | 73.3 | – |
| | Novelty (%) | 99.6 | 100 | 100 | 100 | 100 | 73.3 | 46.7 | 100 | – |
| OTA | Netlisting Validity (%) | 100 | 100 | 0.0 | 100 | 100 | 73.3 | 100 | 96.7 | 93.2 |
| | Simulation Validity (%) | 98.0 | 96.8 | 0.0 | 98.0 | 98.7 | 0.0 | 46.7 | 73.3 | 0.56** |
| | Specification Fulfilling (%) | 82.8 | 68.2 | 0.0 | 0.0 | 65.2 | 0.0 | 6.7 | 43.3 | 0.25** |
| | Novelty (%) | 100 | 100 | 0.0 | 6.1 | 100 | 96.7 | 83.3 | 90.0 | 99.0 |

Table 1: Quantitative comparison of framework performance across three AMS tasks. A vertical rule separates our approach from prior LLM-based methods. * indicates performance with all features included. ** indicates values were taken from a pretrained model. † indicates values from using an LLM backend previously unreported for the framework.

| Framework | Simulator-Driven Training | Technology-Adherent Generation | Expert-Alignment Mechanism | Symmetry Awareness | Syntactically-Correct Generation | Specification-Conditioned Generation |
|-----------|:---:|:---:|:---:|:---:|:---:|:---:|
| AstRL | * | * | * | * | * | * |
| AnalogCoder | | | * | | | * |
| AnalogGenie | | | * | | | * |

Table 2: Qualitative comparison of framework features and properties.

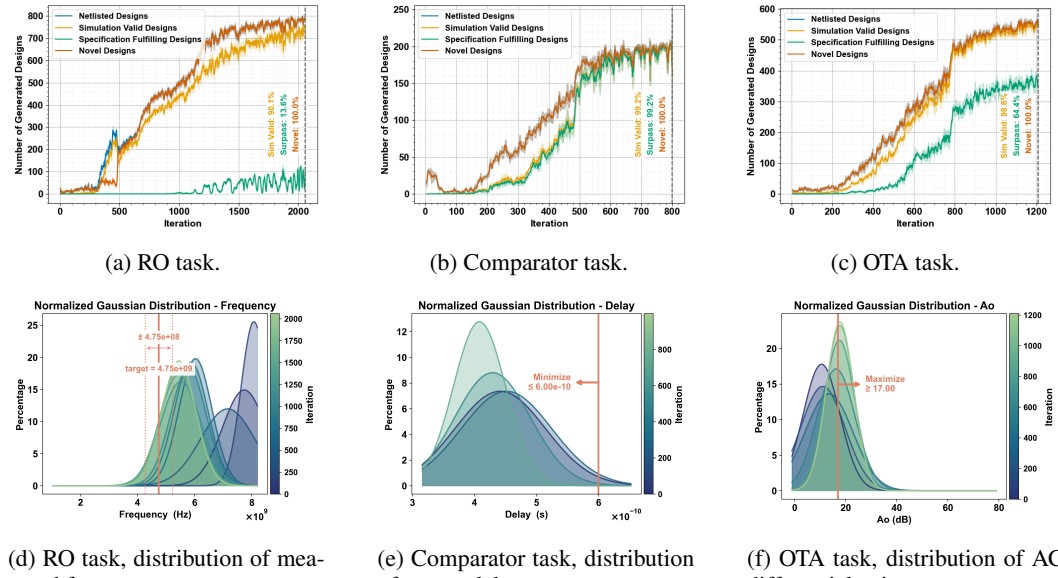

(a) RO task.  (b) Comparator task.  (c) OTA task.

(d) RO task, distribution of measured frequency.  (e) Comparator task, distribution of output delay.  (f) OTA task, distribution of AC differential gain.

Figure 4: Design task outputs. Figs. (a, b, c) show training dynamics and the number of generated designs. Figs. (d, e, f) show the shift of select performance specifications over training iterations for each task using fitted distributions.

## 4.2 DESIGN TASK TRAINING DYNAMICS AND CIRCUIT OPTIMIZATION

A key strength of our simulation-based reinforcement learning formulation is the ability to directly optimize towards desired circuit specifications. For each task, the training dynamics and distributional shifts in circuit performance are shown in Fig. 4.

Figs. 4a, 4b, 4c show the training dynamics for the RO, comparator, and OTA, respectively. Over training iterations, the agent learns to progressively output more designs, as well as improve the performance of the circuits generated. Figs. 4d, 4e, 4f further demonstrate a robust shift in the distribution of measured circuit performance through training progression. Each normalized distribution represents the mean and the variance of the generated designs of a single iteration.

## 4.3 ABLATIONS

For each task, we demonstrate the effect of removing one of four key algorithmic components: (1) the discriminator, (2) behavioral cloning (BC), (3) action masking, and (4) symmetric addition modifiers. No inherent symmetry exists for the RO task, so (4) is not applicable. For the comparator task, AstRL performed effectively without requiring symmetric modifiers, so (4) is not assessed.

These ablation results demonstrate that structural inductive biases and domain constraints are critical to achieving strong generative performance. For circuits lacking inherent symmetry, the BC, action masking, and the discriminator were essential to producing functional circuits. Only when all three mechanisms were present did simulation validity and specification fulfillment rise to performant levels in the RO and comparator benchmarks.

We further note the importance of action space reduction through masking. Despite the trajectory depth reduction afforded by symmetric modifiers, the OTA benchmark only performed reasonably when masking was present. Interestingly, we note that BC enables agents to settle on nominal solutions in challenging design environments by defaulting to closely imitating expert designs.

Finally, we highlight the powerful inductive bias introduced by symmetric action modifiers. Without modifiers, we observe that the framework generates many functional designs for the OTA task but none that fulfill specifications. This suggests that the symmetric modifiers regularize the design

space, enabling tractable optimization without collapses into local optima. The results are summarized in Table 1.

## 4.4 COMPARISONS WITH BASELINES

As previously described, we evaluate our approach against the LLM-based AnalogCoder (Lai et al., 2024) and the transformer-based AnalogGenie (Gao et al., 2025), omitting Chang et al. (2024); Dong et al. (2023) due to scope mismatch or incompatibility with standard simulator-driven evaluation. At time of writing, AnalogCoderPro (Lai et al., 2025) was published, though code was not released.

**Evaluation Consistency:** To ensure consistent and fair comparison, chosen baselines were adapted to include the three described experiments. Importantly, we note several constraints in the evaluation process. In AnalogCoder, a synthetic placeholder process technology (approximately 180 nm) is used to define base components instead of a real foundry-provided process technology (e.g., TSMC, Skywater, Intel). Since this placeholder directly couples into AnalogCoder's prompts and reference libraries, we adjust the task requirements to maintain compatibility.

In AnalogGenie, released implementations do not contain simulator-based verification pipelines for validating design performance. Owing to this, we introduce netlisting utilities to convert the custom Eulerian circuit tokenization scheme into standard netlists used by simulators. Resulting designs were sized and biased using Bayesian Optimization (Balandat et al., 2020), in line with established domain work (Touloupas et al., 2021; Chen et al., 2022). In practice, AnalogGenie produces components that are not simulatable or cannot be fabricated in modern CMOS process technologies (i.e., bipolar junction transistors). These are excluded from evaluations to prevent dilution. AnalogGenie additionally leverages RLHF to align generated designs in the absence of simulations. To enable this, labeled datasets were constructed for each task, though tuning with the released implementation did not yield expected improvements. As a result, we omit reporting for the new RO and comparator tasks and use previously reported values, where applicable, for the OTA task. Table 1 summarizes these results.

**Netlist Validity:** Ensuring that generated circuits map to syntactically valid textual netlists is a prerequisite for meaningful simulator-based evaluation. AnalogCoder achieves an average netlist validity rate of 92.2% and 60.0% with GPT-5 and GPT-3.5 backbones respectively, indicating significant improvement conditioned on underlying LLM performance. With RLHF, AnalogGenie reported a 93.2% netlist validity rate. Our approach achieves 100% validity across all tasks, attributable to our environment formulation that essentially limits the generation of invalid structures.

**Simulation Validity and Specification Fulfillment:** Circuits must converge in simulation and satisfy design specifications. AnalogCoder (GPT-5) achieves an average 75.5% simulation validity rate and 45.5% specification fulfillment rate. As shown in Table 1, performance is strongly correlated with LLM domain reasoning capabilities. AnalogGenie achieved a 0.56% simulation validity rate and a 0.25% specification fulfillment rate on the OTA task when using only the pretrained model. Pretrained model performance is reasonable given the strong representation of amplifier-based designs in the original dataset (56.1% of total elements) when compared to the oscillators (6.96%) and comparators (2.60%). We note that with RLHF tuning, potential performance would be stronger.

Our framework achieves significantly improved performance in simulation validity and specification fulfillment compared to other frameworks. This can be directly attributed to the incorporation of supervised, ground-truth simulator feedback during training. In the absence of reliable proxy predictors for circuit performance, precise and domain-grounded reward signals enable generation towards functionally correct and high-performing circuit topologies.

**Novelty:** AnalogCoder (GPT-5) and AnalogGenie achieve high novelty rates of 96.7% and 99.0% respectively. Our framework also achieves a high novelty rate of 100%. High rates across all methods reflect the expressive capacity of component-level generative models. However, only our framework guarantees structural validity, ensuring that novelty is not achieved at the expense of feasibility.

## 4.5 COMPUTATION COST

The training times for each task can be found in the Table 3. The GPU model used is listed. Only one GPU is used for each task. Circuit simulations were parallelized such that up to 16 simulation jobs could be run at a time on parallel CPU cores:

Table 3: Training performance across design tasks.

| Task | GPU | Time to First Spec-Meeting Design | Time to Convergence | Last / First Iteration Time Ratio |
|------|-----|-----------------------------------|---------------------|-----------------------------------|
| **RO** | RTX 4090 | 15.65 hours | 11.39 days | 6.08 |
| **Comparator** | RTX 4090 | 2.46 hours | 5.67 days | 5.67 |
| **OTA** | L40S | 3.33 minutes | 4.65 days | 4.53 |

The gap between "first valid design" and "full convergence" reflects two use cases: 1) direct optimization of a given topology, where only a single valid design is needed; 2) full topology-space exploration, where convergence is required.

The last/first iteration ratio shows that simulation-verified methods spend most of their time on CPU-based circuit simulation once functional designs are frequently produced. Differences across tasks arise from different testbench complexities, which govern simulation time.

For any method that requires simulation-validated topologies, simulation time fundamentally dominates overall runtime. AstRL's efficiency is therefore in line with the inherent cost of simulator-in-the-loop design exploration, which provides reliable specification satisfaction in the absence of accurate surrogate models.

## 5 CONCLUSION

We have introduced AstRL, a novel simulator-driven, graph-based, reinforcement learning method of generating AMS circuits at the transistor-level. Our method makes several contributions to circuit topology discovery and generation: (1) policy-gradient based formulation of AMS design, (2) generalized reward design applicable across AMS design tasks, (3) expert alignment mechanisms through behavioral cloning, and (4) robust metric definitions and validation processes to ensure practical application to AMS design. We highlight the efficacy of strong inductive biases encoded in our action space and environment design, and note that our approach is applicable to a broad range of AMS circuits spanning analog and digital behaviors. To this end, AstRL achieves 100% valid designs by construction and significantly improved simulation validity and specification fulfillment rates when compared to SoTA frameworks. This constitutes an important step forward towards automated circuit design.

# 6 REPRODUCIBILITY STATEMENT

We provide our dataset, testbench setup scripts for each design task, and the full source code for agent and environment definitions. Due to the requirement of interfacing with commercial simulation tools in a physical process technology found typically within secured environments, certain aspects of the domain-specific environment may be difficult to replicate. To mitigate this, we have included extensive netlist and simulation result artifacts for each design task. These artifacts were used directly in our evaluation of defined metrics. They can be used to reproduce our reported performance, and serve as representative references for future work, including those that may define new or separate metrics.

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

## A  APPENDIX

### A.1  DATASET

Our dataset consists of 1172 circuits, a modified subset of Gao et al. (2025). Circuits were manually verified for supply and ground connectivity to ensure functionality. The distribution of circuit types found in this modified dataset is shown in Fig. 5.

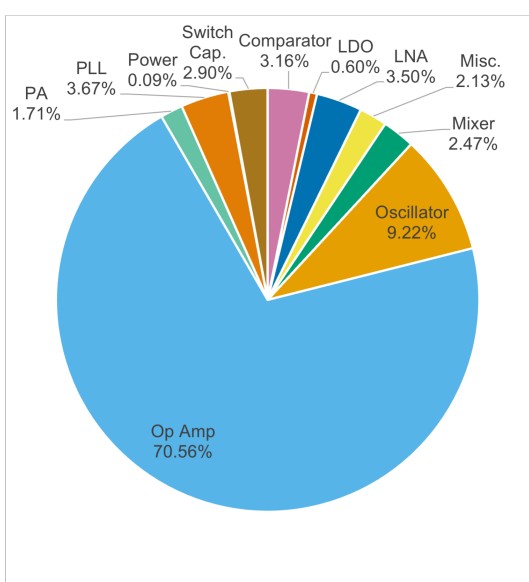

Figure 5: Breakdown of dataset components.

### A.2  SYMMETRIC ACTION MODIFIERS

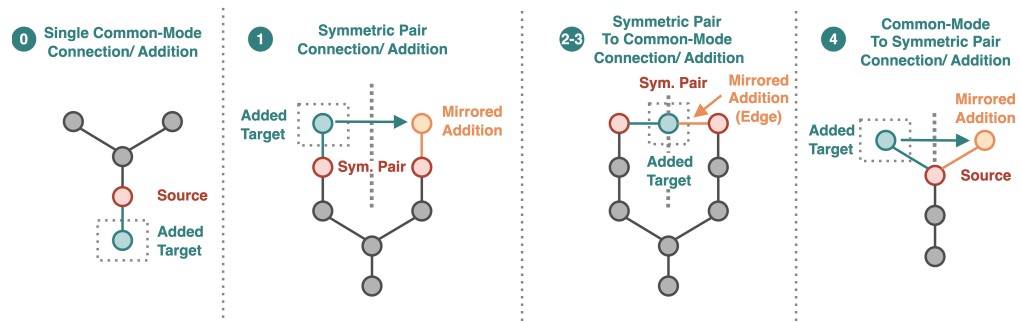

Figure 6: Overview of symmetric action modifiers. For clarity, target nodes are always added.

This section details symmetric modifiers for action subvectors. During graph construction, the environment tracks the addition of new nodes, classifying them into two categories: (1) symmetric and (2) common-mode. Symmetric nodes come in pairs, and are mirrored across the the supply-ground axis. Common-mode nodes lie directly on the supply-ground axis. Nodes are exclusively either common-mode or symmetric. An overview of all five available modifiers is shown in Fig. 6.

**Single Common-Mode Modifier**: Enables a simple connection between a common-mode source node and a common-mode target node.

**Symmetric Pair Modifier**: For an action subvector connecting a symmetric source node to either an existing or newly-added symmetric node, the modifier mirrors the action across the supply–ground axis to its corresponding symmetric counterpart.

**Symmetric Pair to Common-Mode Component Modifier:** For an action subvector connecting a symmetric source node to either an existing or newly-added common-mode component, create a second connection between the common-mode component and the corresponding symmetric counterpart. To satisfy structural validity constraints, the edge attributes of the second edge must be adjusted. For passives, the resulting edge attributes on the two edges must be (P-, P+). For transistors, this must be (D, S).

**Symmetric Pair to Common-Mode Net Modifier:** For an action subvector connecting a symmetric source node to either an existing or newly-added common-mode net, create a second connection between the common-mode net and the corresponding symmetric counterpart. Unlike the prior case, both edges share identical attributes in this scenario.

**Common-Mode to Symmetric Pair Modifier:** For an action subvector connecting a common-mode source node to a newly-added symmetric node, create a second connection between the common-mode source node and the corresponding symmetric counterpart.

## A.3 STRUCTURAL CONSTRAINTS

This section outlines the structural constraints enforced by our environment which are necessary for structural consistency. First, in the action vector definition, $a_{\text{source node}}$ must always be drawn from existing nodes in graph $G$. Then, $a_{\text{target node}}$ can be drawn from either existing nodes, or be added from a scaffold list. Second, every terminal belonging to a component must be connected to a single net to prevent shorts or floats. Third, component nodes must connect to a net node, before connecting to any another component node. This allows net nodes to capture multiple component connectivity. Finally, all components must have fully-connected terminals as a prerequiste for termination.

## A.4 ADDITIONAL RESULTS

For each task, we include examples of representative generated circuits with output waveforms.

### A.4.1 RING OSCILLATOR

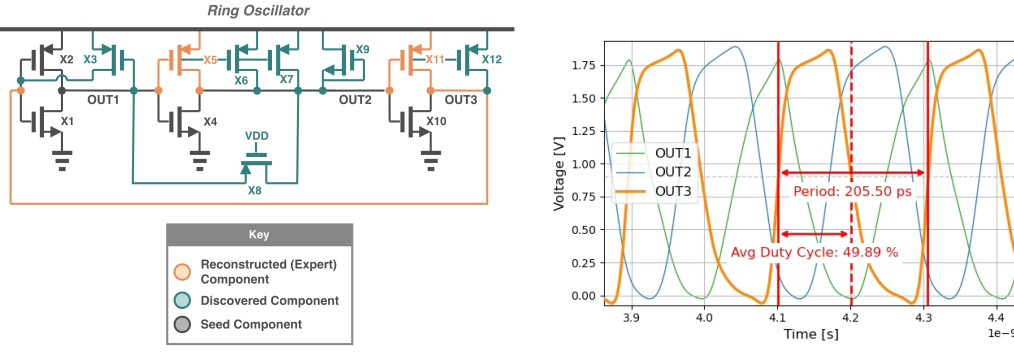

(a) Generated RO example. 4.87 GHz oscillation frequency, 49.9% average duty cycle, 429uW power consumption.

(b) Output transient waveform corresponding to Fig. 7a

Figure 7: Generated RO circuit and corresponding transient response.

### A.4.2 COMPARATOR

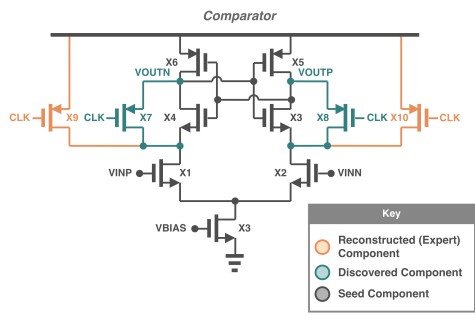

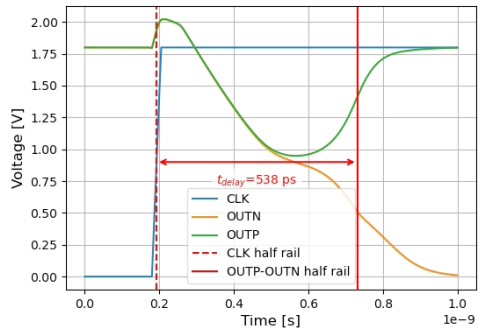

(a) Generated comparator example. 538 ps delay, 0.596mVrms input referred noise, 189uW power consumption.

(b) Output transition transient waveform corresponding to Fig. 8a

Figure 8: Generated comparator circuit and corresponding transient response.

### A.4.3 OPERATIONAL TRANSCONDUCTANCE AMPLIFIER

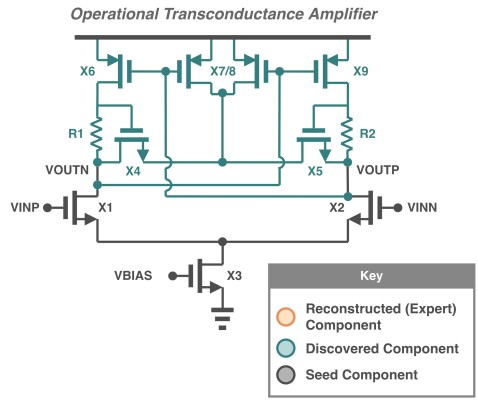

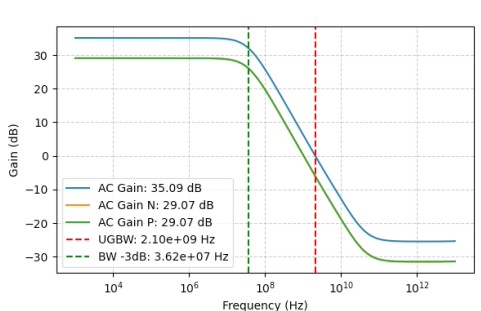

(a) Generated OTA example. 35.09dB AC gain, 36.2 MHz 3dB bandwidth, 700uW power consumption.

(b) AC response corresponding to Fig. 8a. AC gain on the N and P side overlap, showing low gain mismatch.

Figure 9: Generated OTA circuit and corresponding AC response.