# OpenReview forum: "AstRL: Analog and Mixed-Signal Circuit Synthesis with Deep Reinforcement Learning"
_ICLR.cc/2026/Conference — Submitted to ICLR 2026_

### Official Review · Reviewer_XRPA · 2025-10-16

**Soundness:** 2
**Presentation:** 3
**Contribution:** 2
**Rating:** 4
**Confidence:** 3

**Summary:**

This paper proposes AstRL, a deep reinforcement learning approach for automated analog and mixed-signal (AMS) circuit design. The method formulates circuit synthesis as a sequential graph generation problem using PPO with behavioral cloning. Key features include: (1) symmetry-aware action space with structural constraints, (2) simulator-embedded training environment providing ground-truth feedback, and (3) discriminator-based similarity rewards for expert alignment. The approach is evaluated on three circuit design tasks (ring oscillator, comparator, OTA) and achieves 100% netlist validity and >90% simulation validity, outperforming LLM-based baselines.

**Strengths:**

1. Well-structured paper with clear motivation
2.  Action masking and symmetry-aware modifiers ensure 100% valid circuits by construction
3. Combines multiple techniques (graph generation, PPO, BC, discriminator rewards) effectively

**Weaknesses:**

1. Core techniques (graph-based circuit generation, PPO+BC, discriminator rewards) are not new. Main contribution is engineering integration rather than algorithmic innovation.
2. No training time, wall-clock time, or simulation budget reported
3. No justification or ablation on GINE architecture

**Questions:**

1. How is the efficiency of the algorithm with the SPICE simulator in the loop?

---

> ### Author Response · Authors · 2025-11-24
>
> We would like to thank the reviewer for their insightful feedback on the work. We address the primary concerns below.
>
> ### **Novelty** ###
>
> > "Core techniques (graph-based circuit generation, PPO+BC, discriminator rewards) are not new"
>
> AstRL uses PPO and behavioral cloning, but its novelty lies in a **unique problem formulation and reward** that enables transistor-level, specification-driven topological synthesis.
>
> To our knowledge, AstRL is the first RL framework in this domain that:
>
> 1. Operates directly on topology graphs,
> 2. Incorporates expert knowledge into generation,
> 3. Uses simulation-based functional rewards for topology refinement, and
> 4. Directly conditions generation on provided specifications.
>
> Critically, the incorporation of simulator-driven functional feedback for continuous optimization represents a significant step towards **validated and verified** topology generation.
>
> From the domain perspective, similarity-based or proxy metrics are insufficient in analog mixed-signal circuit design, where small structural deviations (e.g., a single missing connection) can result in complete functional failure. This is demonstrated through our comparisons with baselines. Therefore, identifying and applying techniques compatible with simulator feedback represents both a significant challenge and a corresponding contribution.
>
> ### **Training Time and Efficiency** ###
>
> The training times for each task can be found in the table below. The GPU model used is listed. Only one GPU is used for each task. Circuit simulations were parallelized such that up to 16 simulation jobs could be run at a time on parallel CPU cores:
>
> | **Task**     | **GPU** | **Time to First Specification Fulfilling Design** | **Time to Convergence** | **Last vs First Iteration Time Ratio** |
> |--------------|---------|----------------------------------------------------|---------------------------|-----------------------------------------|
> | **RO**        | 4090    | 15.65 hours                                       | 11.39 days                | 6.08                                    |
> | **Comparator**| 4090    | 2.46 hours                                        | 5.67 days                 | 5.67                                    |
> | **OTA**       | L40S    | 3.33 minutes                                      | 4.65 days                 | 4.53                                    |
>
> The gap between “first valid design” and “full convergence” reflects two use cases: 1) direct optimization of a given topology, where only a single valid design is needed; 2) full topology-space characterization, where convergence is required.
>
> **Simulation is a bottleneck common to all existing design processes, and not the result of any architectural decisions.** To ensure that circuits function properly, according to the specifications, they must be simulated. As discussed in the prior section, there are no domain-accepted proxy metrics (to the author's knowledge). The provided points of comparison (AnalogCoder and AnalogGenie) do not report end-to-end simulation-validated runtimes for the same tasks. AnalogCoder is training-free but does not report a simulator-in-the-loop inference cost. A follow-on to AnalogGenie reports <1 hour of RLHF fine-tuning on an A100 [1], but this excludes any circuit simulation.
>
> The last/first iteration ratio shows that simulation-verified methods spend most of their time on CPU-based circuit simulation once the agent begins converging on functional design. Differences across tasks arise from different testbench complexities, which govern simulation time.
>
> We have added section 4.5 to the manuscript to report training cost.
>
> ### **GINE Architecture** ###
>
> We selected GINE because it provides 1-WL expressivity via edge-conditioned message passing [2], which is important for circuits where device types and connection types directly affect functionality. As a result, GINE is well-suited for domain applications, especially when considering the sensitivity of the circuit structure to functionality. The motivation for GINE selection has been further clarified in section 4.1.
>
> [1] Jian Gao et al. EVA: An Efficient and Versatile Generative Engine for Targeted Discovery of Novel Analog Circuits. 2025 62nd ACM/IEEE Design Automation Conference (DAC). 2025, pp. 1–7.
>
> [2] Keyulu Xu et al. “How Powerful are Graph Neural Networks?” In: International Conference on Learning Representations. 2019.

---

> > ### Comment · Reviewer_XRPA · 2025-11-25
> >
> > I acknowledge the author's thoughtful rebuttal. My concerns about GINE Architecture has been addressed. Here are my additional questions.
> > 1. Can you clarify your contribution: Operates directly on topology graphs and Incorporates expert knowledge into generation with more details? Why is it unique from previous work?
> > 2. I appreciate the author's careful evaluation of Training time and efficiency. My fundamental question is, do the proposed techniques within the paper help to make the training and inference more efficient?

---

> > > ### Author Response · Authors · 2025-11-26
> > >
> > > We thank this reviewer for the rapid feedback and continued re-evaluation of the paper. We address each point in turn.
> > >
> > > ###  **Q1** ###
> > > > Operates directly on topology graphs.
> > >
> > > Prior graph domain work [1] operates at the level of **sub-block compositions**. Our work operates at **transistor-level granularity**. AstRL is therefore able to capture interactions (e.g., capacitive loading, impedance interactions) that emerge only when modeling at the granularity of individual devices.
> > >
> > > In contrast, [2],[3],[4] all require sequentializations of the circuit graph. These generative models do not directly act on the graph itself, and permutation invariance is approximated indirectly via data augmentation or sequence design.
> > > In AstRL, we define states and actions on the circuit graph. This combination of transistor-level graph editing and RL with simulator feedback is, to our knowledge, not currently present in other domain work.
> > >
> > > > Incorporates expert knowledge into generation
> > >
> > > Our formulation conditions directly on designer-proposed subcircuits (incomplete subblocks), mirroring typical analog workflows in which engineers start from a partial topology and subsequently refine it. This is markedly different from the largely unconditioned generation of [3] and the commercial-LLM backed approach of [2]. Furthermore, a symmetry-aware action space leverages graph automorphism to construct differential circuits, a mechanism not found in prior sequentialization approaches [2–4].
> > >
> > > Finally, we emphasize that simulation-driven feedback itself is a core contribution. Existing work optimizes proxy or similarity-based metrics that do not guarantee functional behavior. Our agent is trained directly against **simulator-validated performance and specification satisfaction**, closing the gap between generative metrics and actual circuit behavior, and further demonstrating that simulator-in-the-loop topology generation is feasible in practice.
> > >
> > > [1] Zehao Dong, Weidong Cao, Muhan Zhang, Dacheng Tao, Yixin Chen, and Xuan Zhang. CktGNN: Circuit graph neural network for electronic design automation. In *The Eleventh International Conference on Learning Representations*, 2023.
> > >
> > > [2] Yao Lai, Sungyoung Lee, Guojin Chen, Souradip Poddar, Mengkang Hu, David Z. Pan, and Ping Luo. AnalogCoder: Analog circuit design via training-free code generation, 2024.
> > >
> > > [3] Jian Gao, Weidong Cao, Junyi Yang, and Xuan Zhang. Analoggenie: A generative engine for automatic discovery of analog circuit topologies. In *The Thirteenth International Conference on Learning Representations*, 2025.
> > >
> > > [4] Chen-Chia Chang, Yikang Shen, Shaoze Fan, Jing Li, Shun Zhang, Ningyuan Cao, Yiran Chen, and Xin Zhang. Lamagic: Language-model-based topology generation for analog integrated circuits, 2024.
> > >
> > >
> > >
> > > ### **Q2** ###
> > >
> > > > do the proposed techniques within the paper help to make the training and inference more efficient?
> > >
> > > This is an important point, and one that our work addresses to enable feasible simulator-in-the-loop training. Our techniques improve sample efficiency and the effective use of simulation budget, given the invariant cost of a single simulation run. We note three key techniques and their corresponding effects:
> > >
> > > 1.  **Action-Space Masking** reduces the action space to a feasible subset, increasing the proportion of valid rollout steps and improving sampling efficiency. Any remaining invalid actions are detected during per-step verification, ensuring that all final circuits are structurally valid. This prevents unnecessary simulations of nonfunctional structures.
> > >
> > > 2.  **Behavioral Cloning on Expert Subcircuits**  concentrates exploration around realistic regions of the design space. This increases the fraction of meaningful, generated designs (as shown in ablations) and incorporates the information found in expert samples.
> > >
> > > 3.  **Symmetry-Aware Modifiers** reduces rollout depth by applying expert-like edits to differential branches in a single step, lowering the number of environment steps required and further concentrating search in realistic design regions.

---

> > > > ### Comment · Reviewer_XRPA · 2025-11-26
> > > >
> > > > Thanks for the authors' careful response. All my concerns are addressed. I would like to raise my score.

---

> > > > > ### Author Response · Authors · 2025-11-26
> > > > >
> > > > > We thank the reviewer for their helpful feedback and careful evaluation of this work. We appreciate the continued discussion and updated score.

---

### Official Review · Reviewer_VVmR · 2025-10-27

**Soundness:** 3
**Presentation:** 3
**Contribution:** 3
**Rating:** 6
**Confidence:** 4

**Summary:**

This paper proposes AstRL, a deep reinforcement learning (RL)-driven synthesis method for analog and mixed-signal (AMS) integrated circuits (ICs), which corelies in formulating AMS circuit design as a graph generation problem. Built on a policy-gradient framework (Proximal Policy Optimization, PPO), the method leverages Graph Neural Networks (Graph Isomorphism Network with Edge Features, GINE) to enable fine-grained transistor-level topology generation of circuits. It integrates behavioral cloning (for initial expert alignment) and discriminator-based similarity rewards (for trajectory stability maintenance), while incorporating ground-truth feedback from a simulator-embedded environment to optimize circuit performance. Experiments were conducted on three representative design tasks—ring oscillator (RO), comparator, and operational transconductance amplifier (OTA)—using the Skywater 130nm process. Results demonstrate that 100% of generated circuits are structurally valid, and over 90% meet functional requirements, outperforming state-of-the-art baselines such as AnalogCoder (LLM-based) and AnalogGenie (RLHF-based) in terms of simulation validity and specification fulfillment.

**Strengths:**

1.	Notable Originality:
By formulating AMS circuits as a graph generation problem and enabling transistor-level fine-grained generation via RL, the work overcomes limitations of existing LLM-based methods (which lose structural information) and RLHF-based methods (which lack real simulator feedback), establishing a dual-drive mechanism of "expert alignment + exploratory optimization".
2.	Solid Methodological Quality:
Each module—from graph representation (GINE network) and action space (symmetric modifiers + masking) to reward design (three-stage categorized rewards)—is supported by theoretical foundations and validated through ablation studies, without "black-box" design. Experiments use real-world processes and standard tasks, ensuring high result reproducibility.
3.	Strong Domain Significance:
It addresses "structural validity" and "functional compliance" issues of industrial concern. The released code and datasets can drive subsequent research in the field (e.g., multi-module integration of complex AMS circuits and adaptation to larger-scale processes).
4.	High Expressive Clarity:
Framework diagrams, formulas, and experimental tables (multi-task comparison in Table 1) complement each other. The comprehensive overview of related work not only demonstrates an understanding of the field’s history but also clarifies the paper’s positioning, reducing the comprehension burden for reviewers and readers.

**Weaknesses:**

1.	Limited Task Generalization:
Experiments only cover three basic AMS circuits (RO, comparator, OTA) and do not validate more complex industrial-grade circuits (e.g., high-speed SERDES, phase-locked loops (PLLs)). This makes it impossible to determine the method’s adaptability to scenarios with "multi-module coupling and high nonlinearity", restricting the generalization of conclusions.
2.	Lack of Mass Production Metrics:
Existing results focus on "functional correctness" (e.g., frequency, delay, gain) but do not evaluate industrial-critical mass production metrics such as area, power consumption, and yield. The absence of actual tape-out test results for generated circuits makes it difficult to assess the method’s practicality in mass production.
3.	Insufficient Analysis of Expert Data Impact:
Behavioral cloning relies on a dataset of 1,172 expert circuits, but the impact of "expert data scale/diversity" on model performance is not analyzed (e.g., whether the initialization effect of behavioral cloning degrades significantly when data volume is reduced by 50%). This hinders the method’s application in small-data scenarios.
4.	Unmentioned Computational Efficiency:
No indicators of computational efficiency—such as model training time and average time for generating a single circuit—are reported. AMS design requires rapid iteration; if generating one circuit takes several hours, the method may fail to meet industrial rapid-iteration needs, necessitating supplementary efficiency analysis.
5.	Insufficient related work:
More research for circuit synthesis such as [1] [2] should be discussed and compared in related work or experiments.

[1]. Bai Y, Wang J, Chen L, et al. A Graph Enhanced Symbolic Discovery Framework For Efficient Logic Optimization. The Thirteenth International Conference on Learning Representations.

[2]. Wang Z, Wang J, Yang Q, et al. Towards next-generation logic synthesis: A scalable neural circuit generation framework. Advances in Neural Information Processing Systems, 2024, 37: 99202-99231.

**Questions:**

1.	Has your team attempted to apply AstRL to more complex industrial-grade AMS circuits (e.g., PLLs, SERDES)? If not, could you analyze potential challenges (such as increased trajectory depth and sparse reward signals) that the method may face in scenarios with "multi-module coupling and high nonlinearity"?
2.	The paper does not evaluate mass production metrics (area, power consumption, yield) of generated circuits. Do you plan to supplement tape-out tests or related simulations in the future? These metrics are critical for industrial deployment—could you explain the potential of the existing method in optimizing mass production metrics?
3.	For the dataset of 1,172 expert circuits used in behavioral cloning, how do the scale and diversity of the data impact model performance? If the volume of expert data is reduced (e.g., to only 100 circuits), will the initialization effect and final performance of the model degrade significantly?
4.	When comparing with AnalogGenie, your team supplemented "netlist conversion + Bayesian optimization" for simulator validation. Were the parameter settings of these two steps (e.g., number of iterations and hyperparameters for Bayesian optimization) consistent with the original AnalogGenie method? If there are differences, could they affect the fairness of the comparison results?
5.	What is the computational efficiency of the model? For example, how long does it take to train the model for the three tasks on a single GPU? What is the average time required to generate an OTA circuit that meets specifications? Could you compare its efficiency with existing methods (e.g., AnalogCoder)?

---

> ### Author Response · Authors · 2025-11-24
> **Rebuttal (1/2)**
>
> We appreciate the reviewer’s feedback and perspectives on the work. We have included the additional related work in the updated manuscript in Section 2. We would also like to address the questions raised.
>
> ### **Q1** ###
> >“Has your team attempted to apply AstRL to more complex industrial-grade AMS circuits (e.g., PLLs, SERDES)? If not, could you analyze potential challenges (such as increased trajectory depth and sparse reward signals) that the method may face in scenarios with "multi-module coupling and high nonlinearity"?"
>
> AstRL targets block-level analog topology synthesis, consistent with how complex AMS systems are designed in practice. PLLs and SERDES are compositions of smaller blocks (oscillators, amplifiers, filters, comparators, bias cells, etc.) typically containing tens of devices each. AstRL is designed for these block-level units. Extending RL to enable multi-module architectural exploration is an important and exciting direction, but outside the current scope of this work.
>
>
> ### **Q2** ###
> >“The paper does not evaluate mass production metrics (area, power consumption, yield) of generated circuits. Do you plan to supplement tape-out tests or related simulations in the future? These metrics are critical for industrial deployment—could you explain the potential of the existing method in optimizing mass production metrics?”
>
> Area, power, and statistical yield can be evaluated via post-layout estimation and Monte Carlo simulations. Our current experiments use functional simulation metrics to evaluate results associated with each topology. Incorporating area/power/yield terms directly into the RL reward is conceptually straightforward and compatible with our framework. This would be an interesting next direction to pursue.
>
> ### **Q3** ###
> >“For the dataset of 1,172 expert circuits used in behavioral cloning, how do the scale and diversity of the data impact model performance? If the volume of expert data is reduced (e.g., to only 100 circuits), will the initialization effect and final performance of the model degrade significantly?”
>
> The 1,172 expert circuits are used **only for pretraining**. During RL policy optimization, behavioral cloning uses **only 1–3 expert circuits** relevant to each task. Given the limited number of expert RO and comparator circuits in the pretraining set, as well as the small number of circuits used in task-specific training, RO and comparator results actually **demonstrate that the method is effective in low-data settings**.

---

> ### Author Response · Authors · 2025-11-24
> **Rebuttal (2/2)**
>
> ### **Q4** ###
> > When comparing with AnalogGenie, your team supplemented "netlist conversion + Bayesian optimization" for simulator validation. Were the parameter settings of these two steps (e.g., number of iterations and hyperparameters for Bayesian optimization) consistent with the original AnalogGenie method? If there are differences, could they affect the fairness of the comparison results?
>
> We followed the publicly released AnalogGenie repository, which currently does not include RLHF or netlisting conversion (see open Github issues on https://github.com/xz-group/AnalogGenie/issues). As noted in the paper, we implemented netlist conversion following the tokenization and Eulerian path format described in their documentation to enable simulator-driven verification.
>
> AnalogGenie uses a GA-based sizing backend, but the released repository does not include the GA implementation described in the paper. We therefore used Bayesian optimization as the sizing backend, which is a standard method in analog automation and one highlighted by the same authors as effective (i.e., [1]). Section 4.4 of the paper notes this and emphasizes that RLHF fine-tuning could further improve AnalogGenie’s results.
>
> ### **Q5** ###
> > What is the computational efficiency of the model? For example, how long does it take to train the model for the three tasks on a single GPU? What is the average time required to generate an OTA circuit that meets specifications? Could you compare its efficiency with existing methods (e.g., AnalogCoder)?
>
> The training times for each task can be found in the table below. The GPU model used is listed. Only one GPU is used for each task. Circuit simulations were parallelized such that up to 16 simulation jobs could be run at a time on parallel CPU cores:
>
> | **Task**     | **GPU** | **Time to First Specification Fulfilling Design** | **Time to Convergence** | **Last vs First Iteration Time Ratio** |
> |--------------|---------|----------------------------------------------------|---------------------------|-----------------------------------------|
> | **RO**        | 4090    | 15.65 hours                                       | 11.39 days                | 6.08                                    |
> | **Comparator**| 4090    | 2.46 hours                                        | 5.67 days                 | 5.67                                    |
> | **OTA**       | L40S    | 3.33 minutes                                      | 4.65 days                 | 4.53                                    |
>
> The gap between “first valid design” and “full convergence” reflects two use cases: 1) direct optimization of a given topology, where only a single valid design is needed; 2) full topology-space characterization, where convergence is required.
>
> **Simulation is a bottleneck common to all existing design processes, and not the result of any architectural decisions.** To ensure that circuits function properly, according to the specifications, they must be simulated. To the author's knowledge, there are no domain-accepted proxy metrics. The provided points of comparison (AnalogCoder and AnalogGenie) do not report end-to-end simulation-validated runtimes for the same tasks. AnalogCoder is training-free but does not report a simulator-in-the-loop inference cost. A follow-on to AnalogGenie reports <1 hour of RLHF fine-tuning on A100 [2], but this excludes any circuit simulation.
>
> The last/first iteration ratio shows that simulation-verified methods spend most of their time on CPU-based circuit simulation once the agent begins converging on functional designs. Differences across tasks arise from different testbench complexities, which govern simulation time.
>
> We have added section 4.5 to the manuscript to report training cost.
>
> [1] Jian Gao, Weidong Cao and Xuan Zhang. RoSE: Robust Analog Circuit Parameter Optimization with Sampling-Efficient Reinforcement Learning. In *2023 60th ACM/IEEE Design Automation Conference (DAC)*, 2023, pp. 1-6
>
> [2] Jian Gao et al. EVA: An Efficient and Versatile Generative Engine for Targeted Discovery of Novel Analog Circuits. 2025 62nd ACM/IEEE Design Automation Conference (DAC). 2025, pp. 1–7.

---

### Official Review · Reviewer_MJxH · 2025-10-30

**Soundness:** 2
**Presentation:** 2
**Contribution:** 2
**Rating:** 2
**Confidence:** 5

**Summary:**

The paper proposes an RL-based method to generate optimized analog circuits that meet given targets. Three downstream tasks are evaluated to show the performance of the proposed RL method.

**Strengths:**

Leveraging RL to search optimal-sized circuit topologies is exciting.

The circuit-level graph modeling by considering rich domain knowledge is well presented.

The RL framework is also introduced well, with detailed reward design, which is the key to this framework.

**Weaknesses:**

The scalability of the method is limited, especially when the number of devices in analog circuits increases.

The training of the method is very time-consuming. There are two searching loops in the framework. First, the RL will search topologies, and meanwhile, the reward generation needs to search for optimal device parameters. The latter could be even time-consuming. This makes the framework unlikely to be practical.

Evaluations and comparisons are pretty unfair. The three circuits for evaluations are quite small compared to the circuits in Analogcoder and AnalogGenie. If you focus solely on these three circuits and conclude that your RL method achieves the state-of-the-art (SOTA), it is not convincing.

**Questions:**

How is each circuit designed? Is a separate agent required to be trained to achieve this purpose?

What do the different normal distribution curves represent in Figure 4d, e, f?

What is the training cost?

What is the initial starting point of each episode in your RL agent, and how do you determine the end of each episode?

If you just use this RL to achieve the given design target, why not pick up an existing topology?

**Details Of Ethics Concerns:**

N/A.

---

> ### Author Response · Authors · 2025-11-24
> **Rebuttal (1/2)**
>
> We’d like to thank the reviewer for their feedback and insights. We would like to address some of the points discussed below.
>
> ### **General** ###
>
> >“There are two searching loops in the framework”:
>
> AstRL contains a single RL loop, as shown in Fig. 1. Device parameter assignment is not an inner optimization loop; all components use unit-sized devices, as typical in modern, scaled technologies, and multiplicity is generated via parallel connections. This has been further clarified in Section 4.1.
>
> >“The scalability of the method is limited”:
>
> AstRL targets block-level analog topology synthesis, consistent with how complex AMS systems are designed in practice. Practical extension to large circuits should involve multi-module compositions. This is an important direction, but ultimately is a separate design problem.
>
> >“Evaluations and comparisons are pretty unfair”:
>
> We followed standard evaluation practice by generating three shared, reproducible, executable tasks implemented across frameworks for fair comparison of each framework’s design capability. We highlight that generating large circuits is only one aspect of achieving SoTA performance. This is also observed in the original AnalogCoder evaluation that splits design tasks by varying difficulty levels. We would like to emphasize that the use of **gradient-based methods to propagate ground-truth simulation feedback to produce circuits that function according to a real design specification is a crucial feature.**

---

> ### Author Response · Authors · 2025-11-24
> **Rebuttal (2/2)**
>
> We further address the questions raised:
>
> ### **Q1** ###
> > “How is each circuit designed? Is a separate agent required to be trained to achieve this purpose?”
>
> Each design task specifies a set of performance targets and provides an initial seed topology (Appendix A.4). A separate RL agent is trained per task because the objectives, topology space, and optimality criteria differ across circuit classes. As is analogous to industry teams, separate designers are responsible for separate blocks.
>
> ### **Q2** ###
> >“What do the different normal distribution curves represent in Figure 4d, e, f?”
>
> Figs. 4d, 4e, 4f further demonstrate the measured circuit performance through training iterations. Darker curves are for earlier training iterations. More green curves are for later training iterations. From the designs generated at each iteration, a mean and standard deviation are fit. This has been further clarified in section 4.2.
>
> **Q3**
> >“What is the training cost?”
>
>
> The training times for each task can be found in the table below. The GPU model used is listed. Only one GPU is used for each task. Circuit simulations were parallelized such that up to 16 simulation jobs could be run at a time on parallel CPU cores:
>
> | **Task**     | **GPU** | **Time to First Specification Fulfilling Design** | **Time to Convergence** | **Last vs First Iteration Time Ratio** |
> |--------------|---------|----------------------------------------------------|---------------------------|-----------------------------------------|
> | **RO**        | 4090    | 15.65 hours                                       | 11.39 days                | 6.08                                    |
> | **Comparator**| 4090    | 2.46 hours                                        | 5.67 days                 | 5.67                                    |
> | **OTA**       | L40S    | 3.33 minutes                                      | 4.65 days                 | 4.53                                    |
>
> The gap between “first valid design” and “full convergence” reflects two use cases: 1) direct optimization of a given topology, where only a single valid design is needed; 2) full topology-space characterization, where convergence is required.
>
> **Simulation is a bottleneck common to all existing design processes, and not the result of any architectural decisions.** To ensure that circuits function properly, according to the specifications, they must be simulated. To the author's knowledge, there are no domain-accepted proxy metrics. The provided points of comparison (AnalogCoder and AnalogGenie) do not report end-to-end simulation-validated runtimes for the same tasks. AnalogCoder is training-free but does not report a simulator-in-the-loop inference cost. A follow-on to AnalogGenie reports <1 hour of RLHF fine-tuning on an A100 [1], but this excludes any circuit simulation.
>
> The last/first iteration ratio shows that simulation-verified methods spend most of their time on CPU-based circuit simulation once the agent begins converging on functional designs. Differences across tasks arise from different testbench complexities, which govern simulation time.
>
> We have added section 4.5 to the manuscript to report training cost.
>
> **Q4**
> >“What is the initial starting point of each episode in your RL agent, and how do you determine the end of each episode?”
>
> The initial starting points are shown in the Appendix A.4 as the black-colored transistors. The end of each episode is learned by the RL agent.
>
> **Q5**
> >“If you just use this RL to achieve the given design target, why not pick up an existing topology?”
>
> Existing topologies may not fundamentally be able to hit a performance point without topological modification - our approach **fundamentally enables performance-aware topology synthesis.**
>
> [1] Jian Gao et al. EVA: An Efficient and Versatile Generative Engine for Targeted Discovery of Novel Analog Circuits. 2025 62nd ACM/IEEE Design Automation Conference (DAC). 2025, pp. 1–7.

---

### Meta-Review · Area_Chair_UCEj · 2026-01-03

**Summary:**

This paper proposes AstRL, a simulator-in-the-loop deep RL framework that formulates circuit synthesis as a transistor-level graph generation problem using PPO, with strong inductive biases and optional expert alignment. The evaluation on three tasks reports high structural validity and strong simulator-validated performance relative to selected baselines.

The reviewers’ discussion centers on (i) whether the contribution is primarily an engineering integration of known components rather than a clear algorithmic advance, (ii) limited empirical scope (only three relatively small circuits/tasks) and lack of evidence for scalability/generalization to more complex industrial circuits, and (iii) practical feasibility given heavy simulator cost and unclear end-to-end efficiency and fair baseline reproduction. While the rebuttal clarifies some misunderstandings and adds training-time reporting, key concerns about scalability, baseline comparability, and real-world practicality remain insufficiently resolved. Thus I recommend Reject.

**Reviewer Concerns:**

Concerns addressed by rebuttal / discussion:
- Misunderstanding about two search loops (MJxH): Authors clarified there is a single RL loop and that device sizing is not an inner optimization loop in their framework.

- Computational cost transparency (VVmR, XRPA, MJxH): Authors provided wall-clock training time statistics per task and emphasized simulator cost as the bottleneck. This improves transparency relative to the original submission.

- GNN choice justification (XRPA): The motivation for selecting GINE and its relevance to edge-conditioned circuit structure was clarified, and the reviewer indicated this concern was resolved.

- Positioning as block-level synthesis (VVmR): Authors clarified scope: targeting transistor-level sub-blocks rather than full-system optimization (e.g., PLL/SERDES), and that extension to multi-module exploration is future work.

Concerns still outstanding / only partially addressed:

- Scalability & generalization evidence (MJxH, VVmR): The empirical validation remains limited to three tasks, with no convincing demonstration that the approach scales to more complex circuits or multi-module coupling. The “out of scope” response does not substitute for evidence supporting broader claims.

- Practicality and efficiency (MJxH, VVmR, XRPA): Even with added training-time reporting, the reported costs (days of simulator-in-the-loop training to converge) raise concerns about feasibility for rapid design iteration. The rebuttal frames simulation as unavoidable but does not sufficiently demonstrate that the proposed techniques materially improve end-to-end practical usability versus alternatives.

- Baseline comparability and strength of SOTA claim (MJxH, VVmR): The baseline situation remains imperfect: for AnalogGenie, missing public components required the authors to implement netlisting and replace the described sizing backend (GA) with Bayesian optimization. Even if reasonable, these substitutions introduce uncertainty in fairness and make strong SOTA-style conclusions less convincing.

- Novelty bar (XRPA, partially VVmR): While the authors argue novelty via specification-conditioned, simulator-verified, transistor-level graph editing,” the core algorithmic components (PPO+BC, discriminator rewards, constrained action spaces) are established. The work reads more as a system/engineering contribution, and the submission does not yet provide sufficiently compelling new algorithmic insights to meet the venue bar given the above empirical risks.

**Reviewer Scores:**

Reviewer MJxH (2 → 2),
Reviewer VVmR (6 → 6),
Reviewer XRPA (4 → 6).

---

### Decision · Program_Chairs · 2026-01-26

Reject